# Lost in Translation: On the Idiosyncratic Gap between Image Captioning and Generation Models

## Abstract

In this work, we study idiosyncrasies in the caption models and their downstream impact on text-to-image models. We design a systematic analysis: given either a generated caption or the corresponding image, we train neural networks to predict the originating caption model. Our results show that text classification yields very high accuracy (99.70%), indicating that captioning models embed distinctive stylistic signatures. In contrast, these signatures largely disappear in the generated images, with classification accuracy dropping to at most 50% even for the state-of-the-art Flux model. To better understand this cross-modal discrepancy, we further analyze the data and find that the generated images fail to preserve key variations present in captions, such as differences in the level of detail, emphasis on color and texture, and the distribution of objects within a scene. Overall, our classification-based framework provides a novel methodology for quantifying both the stylistic idiosyncrasies of caption models and the prompt-following ability of text-to-image systems.

## 1 Introduction

Synthetic data now plays a central role in training and scaling multimodal systems (Brack et al., 2025; Hammoud et al., 2024; Lai et al., 2024). In state-of-the-art image generation pipelines (e.g., DALL·E 3 (Betker et al., 2024), Playground v3 (Liu et al., 2024), Qwen-image (Wu et al., 2025)), model-generated captions are used to expand training corpora and to refine text–image alignment. This practice implicitly assumes that captions are (i) stylistically neutral or at least interchangeable across captioning models, and (ii) faithfully convertible into visual content by text-to-image (T2I) models. Both assumptions are under-examined.

A growing body of work shows that language models imprint stable, model-specific "fingerprints" that enable source attribution from text (Geva et al., 2021; McGovern et al., 2025; Wanli et al., 2025; Sun et al., 2025). Similar dataset/model signatures have been reported in vision (Torralba & Efros, 2011; Corvi et al., 2023; You et al., 2025; Liu & He, 2025). However, it remains unclear whether *caption-level* idiosyncrasies produced by vision–language models (VLMs/MLLMs) (Anthropic, 2024; OpenAI, 2022; Liu et al., 2023; Google, 2023) propagate into the *images* produced by downstream T2I systems. If such cross-modal transfer is weak, synthetic-caption pipelines could quietly introduce distributional biases at the text stage that do not materialize visually, complicating the use of captions as faithful supervisory signals.

We investigate this question with a simple, model-agnostic methodology: "name-that-model" classifiers on both sides of the caption → image interface. Given an image and prompt, multiple captioning models produce captions; we first train a text classifier to attribute each caption to its source model. We then feed those same captions into a fixed T2I model and train an image classifier to attribute the *generated images* to the caption source. If caption idiosyncrasies reliably transfer across modalities, attribution should remain high in the image domain; if not, we obtain a direct, quantitative, and interpretable measure of a cross-modal "translation gap."

Empirically, we find that caption idiosyncrasies are highly pronounced in text but largely *dissipate* when transferred into images. On 30k captions per model spanning diverse image sets, a straight-forward BERT-based classifier achieves 99.70% accuracy in identifying the captioning model. In

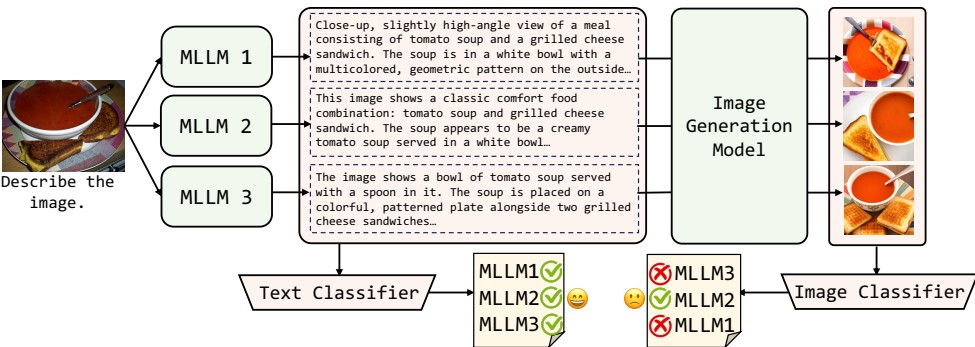

Figure 1: Overview of our pipeline. MLLMs generate captions for an image; a text classifier attributes captions to their source model, and an image classifier attempts to do the same on images generated from those captions.

contrast, after rendering those captions with modern T2I systems, image-space attribution drops substantially, peaking at only $\sim \mathbf{50}\%$ with the current best model (Flux-schnell (Labs et al., 2025)), which is only modestly above the 33.3% chance level for three classes. As a strong reference point, the identical image classifier reaches $\sim 76.7\%$ accuracy when distinguishing *natural* image sources of comparable scale, underscoring that the difficulty is specific to generated images rather than to the classifier.

To understand why, we analyze linguistic and content features of captions. TF–IDF phrase statistics, color/texture vocabularies, and compositional terminology reveal stable, model-specific preferences (e.g., viewpoint/angle wording, ambience/lighting emphasis, or concise compositional framing). We find that paraphrasing the caption still preserve $> 95\%$ attribution accuracy. It is a strong indication that the fingerprints extend beyond surface phrasing to choices about *what* to describe and *how* to structure it. Yet many of these choices fail to manifest reliably in the generated images, especially along axes such as level of detail, nuanced color/texture, and object layout.

These results provide an operational measurement of a cross-modal idiosyncratic gap: stylistic and content-selection signals that are strong in captions are not faithfully realized by current T2I models. Practically, this suggests that (i) aggregating captions from heterogeneous captioners may inject text-domain biases that do not become visual supervision, and (ii) instruction-following in T2I remains a key bottleneck for transferring caption semantics beyond object keywords.

In summary, our work makes the following contributions:

• We propose a simple, scalable attribution framework that quantifies model-specific idiosyncrasies on both captions and the images generated from them.

• We demonstrate that source attribution of captions is nearly perfect (99.70%), whereas attribution based on the corresponding generated images is markedly weaker (peaking at $\sim 50\%$), thereby exposing a substantial cross-modal "translation gap."

• Through lexical and structural analyses (TF–IDF phrases, color/texture vocabularies, composition terms) and paraphrasing robustness, we trace caption fingerprints to deeper content-selection and perspective patterns that current T2I models fail to preserve.

• We propose attribution-as-evaluation as a complementary metric for prompt-following: instruction following should increase transfer of caption idiosyncrasies into images, narrowing the gap.

## 2 BACKGROUND

### 2.1 IDIOSYNCRASIES IN LARGE LANGUAGE MODELS

Large language models achieve remarkable performance across diverse tasks by leveraging the statistical and semantic regularities embedded in large-scale corpora. Beyond their generalization capabilities, recent studies reveal that LLMs also exhibit stable, model-specific *idiosyncrasies* in

generated text. These idiosyncrasies, expressed as consistent stylistic and distributional patterns, act as implicit fingerprints that make outputs attributable to their source models (Geva et al., 2021; McGovern et al., 2025; Wanli et al., 2025). Building on this view, Sun et al. (2025) formalize an attribution task where a classifier predicts the source model from generated samples, and show that such fingerprints persist across model families and prompting conditions, suggesting that differences extend beyond surface token statistics. Similar findings appear in authorship attribution and neural text forensics (Uchendu et al., 2020; Antoun et al., 2023; Dunlap et al., 2025), where stylistic or distributional features reveal model origin even under paraphrasing or translation. These observations naturally raise the question of whether analogous signatures also arise in multimodal settings.

## 2.2 Idiosyncrasies in Vision and Vision Language Models

Idiosyncratic signatures are not confined to text. In computer vision, classic studies showed that simple classifiers can reliably distinguish between datasets, revealing systematic biases beyond semantic content (Torralba & Efros, 2011; Corvi et al., 2023). Similar effects are observed in generative models: diffusion- and GAN-based systems often imprint dataset- or model-specific artifacts that enable reliable attribution (You et al., 2025; Mansour & Heckel, 2024; Tang et al., 2024). These results suggest that high-dimensional visual data still carries persistent distributional cues.

With the rise of vision–language models (VLMs), such concerns extend across modalities. Models like CLIP learn joint embeddings of text and vision (Radford et al., 2021), yet their outputs may still reflect stylistic biases inherited from training. Recent work highlights that VLMs, when used for captioning or generation, can produce model-specific vocabulary, style, or narrative emphasis (Sun et al., 2025; Dunlap et al., 2025). What remains unclear is whether these linguistic fingerprints propagate across modalities, that is, from captions into the images synthesized by downstream text-to-image systems. This question motivates our analysis.

## 3 Idiosyncrasies in generated image captions

Prior research has demonstrated that large language models exhibit model-specific idiosyncrasies in their outputs. In this work, we ask *does this observation apply to MLLMs and their downstream application, such as captioning?*.

### 3.1 Experimental Setup

To investigate the idiosyncrasies of captions generated by different MLLMs, we formulated a classification task. Given a prompt $p \in \mathcal{P}$ and an image $x \in \mathcal{X}$, each model $M_k$ produces a caption $c = M_k(p, x)$, where $\mathcal{C}_k$ is the set of captions from $M_k$. Each caption $c_i \in \mathcal{C}_k$ is paired with a label $y_i = k$, indicating its source model. For $K$ MLLMs, we train a $K$-way classifier to predict $y_i$ from $c_i$. If caption distributions overlap heavily, accuracy should approach random guessing $(1/K)$; substantially higher accuracy indicates model-specific linguistic fingerprints.

We construct our image pool from several widely used datasets. Specifically, we sample 10,000 images in total: 3,000 each from the validation sets of CC3M (Sharma et al., 2018), COCO (Lin et al., 2014), and ImageNet (Deng et al., 2009), plus 1,000 from MNIST (Deng, 2012).

For caption generation, we employ 3 proprietary MLLMs: Claude 3.5 Sonnet (Anthropic, 2024), Gemini 1.5 Pro (Team et al., 2024), and GPT-4o (Hurst et al., 2024), all accessed via their official APIs. To capture linguistic diversity and range, we design three progressively detailed prompts for every image. These prompts elicit different levels of granularity and complexity, enabling a systematic comparison of captioning styles, lexical choices, and narrative depth across models under uniform prompting conditions. Specifically, the 3 prompts are as follows:

**Coarse captioning prompt:**

> Describe the image.

**Detailed captioning prompt:**

Table 1: Test accuracy of caption classification

| Overall | Accuracy on Claude | Accuracy on Gemini | Accuracy on GPT |
|---------|--------------------|--------------------|-----------------|
| 99.80 | 99.68 | 99.82 | 99.90 |

> Write a detailed caption for the image.

**Very detailed captioning prompt:**

> Tell me everything you can see in the image, including as many visible elements as possible.

We set the maximum output length to 1024 tokens for Prompts 1 and 2, and 4096 for Prompt 3, enabling more detailed descriptions. The generation temperature was fixed at 1.0 across all prompts and models to balance variation and determinism. Each model produced 30k captions (3 prompts per image for 10k images). Captions were split 80/20 at the image level, yielding 72k training and 18k test samples, with all prompts of the same image allocated consistently. For classification, we fine-tuned `BERT-base-uncased` (Devlin et al., 2019) with a [CLS]-based linear head to predict the generating model. Training used the AdamW optimizer with a learning rate of $2 \times 10^{-5}$, weight decay 0.01, batch size 32, 3 epochs, and dropout $p = 0.1$, with a linear decay schedule.

## 3.2 MODEL-SPECIFIC FINGERPRINTS ARE SYSTEMATIC IN CAPTIONS

As shown in Table 1, the classifier achieves an overall accuracy of **99.8%**, far above the random baseline of 33.3%. This near-perfect performance indicates that captions from different MLLMs contain highly distinctive linguistic signals, despite being generated under identical image and prompt conditions. In other words, outputs from `Claude-3.5-Sonnet`, `Gemini-1.5-Pro`, and `GPT-4o` exhibit consistent stylistic or lexical fingerprints that enable reliable attribution.

Per-class accuracies are also uniformly high, each exceeding 99.6%. This suggests that fingerprints are not confined to a single model but are shared across all three, confirming that stylistic biases are a systematic property of MLLM captioning rather than an isolated artifact.

## 3.3 WORD DISTRIBUTION ANALYSIS

We next analyze word distributions to understand what makes captions from different models so separable. We apply TF–IDF (Sparck Jones, 1972) ranking of 2-grams and 3-grams to the generated captions. For each model, we compute the top-10 highest-scoring phrases and list them in Table 2, which provides representative lexical patterns. The results reveal clear biases: `Claude-3.5-Sonnet` frequently emphasizes lighting and visibility (e.g., "lighting suggests," "black," "white"), `Gemini-1.5-Pro` highlights perspective and resolution (e.g., "slightly low angle," "impression," "partially visible"), while `GPT-4o` favors categorical or structural terms (e.g., "image depicts," "feature," "wall"). These tendencies reflect stable narrative preferences: Claude focuses on ambience, Gemini emphasizes viewpoint, and GPT-4o prioritizes compositional framing. Wordclouds (Figure 2) further confirm these stylistic differences.

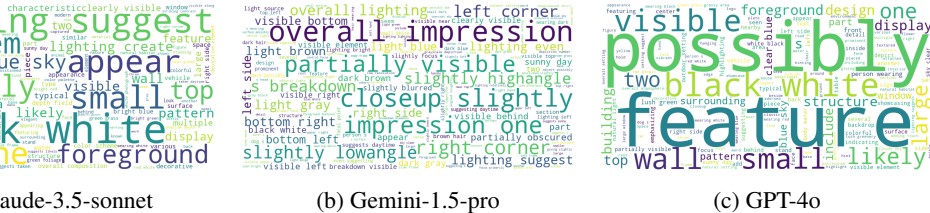

(a) Claude-3.5-sonnet      (b) Gemini-1.5-pro      (c) GPT-4o

Figure 2: Wordclouds of model-generated captions

Table 2: Top distinctive TF-IDF phrases from each model (excluding generic phrases)

| Rank | Claude-3.5-Sonnet | Gemini-1.5-Pro | GPT-4o |
|------|-------------------|----------------|--------|
| 1 | lighting suggests | overall impression | image depicts |
| 2 | visible background | low resolution | image features |
| 3 | light colored | slightly low angle | person wearing |
| 4 | appears taken | close slightly low | handwritten digit |
| 5 | photo taken | eye level | shows person |
| 6 | lighting creates | black background | number black |
| 7 | depth field | high angle view | white black background |
| 8 | background appears | light gray | image shows handwritten |
| 9 | clearly visible | partially visible | setting appears |
| 10 | blue sky | low angle view | clear blue |

These findings show that the three models adopt stable yet distinct descriptive strategies: Claude-3.5-Sonnet foregrounds ambience and lighting, Gemini-1.5-Pro emphasizes viewpoint and framing, and GPT-4o favors categorical and structural terms. Such consistent preferences extend beyond individual words, forming recognizable linguistic fingerprints that explain the high classification accuracy. They further suggest that MLLMs inject narrative biases into captions, which may shape how downstream systems interpret the same images (Section 4).

## 4 IMAGE GENERATION WITH STYLISH CAPTIONS

Given the strong idiosyncrasies observed in captions, a natural question is whether these stylistic signals transfer into the images generated from them. *Do captions from different MLLMs yield visually distinctive images, or do generative models normalize such differences?*

### 4.1 EXPERIMENTAL SETUP

We adopt a parallel experimental setup on the image side. Fixing a text-to-image generator $G$, we use captions $c_i \in \mathcal{C}_k$ produced by different captioning models $M_k \in \{M_1, M_2, \ldots, M_K\}$ as input. This produces generated images

$$\hat{x}_i = G(c_i), \quad \text{with label } y_i = k \text{ if } c_i \in \mathcal{C}_k.$$

We then train an $N$-way classifier over the generated images $\hat{x}_i$ to predict the originating captioning model $M_k$. As in the text domain, classification accuracy above random chance ($1/N$) would indicate that model-specific idiosyncrasies persist in the generated images.

For image-side experiments, we use captions from Claude 3.5 Sonnet, Gemini 1.5 Pro, and GPT-4o, and render them with several widely used T2I systems: Stable Diffusion v1.5 (Rombach et al., 2022), Stable Diffusion v2.1 (Podell et al., 2023), Stable Diffusion XL (Podell et al., 2023), and FLUX.1-schnell (Esser et al., 2024). Following Liu & He (2025), we adopt comparable training settings for the image classifier. Specifically, we use a ResNet-18 backbone trained for 300 epochs with a batch size of 64, the AdamW optimizer (learning rate $5 \times 10^{-4}$, weight decay 0.05), and standard augmentation including Mixup (Zhang et al., 2017) ($\alpha = 0.8$) and CutMix (Yun et al., 2019) ($\alpha = 1.0$). We also apply label smoothing (0.1) and a warmup schedule of 20 epochs.

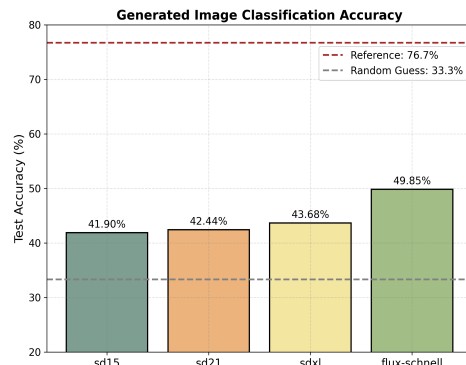

Figure 3: The test accuracy of image classification on generated images is 49.85% with the SOTA model Flux-schnell, whereas classification on a natural image dataset (Liu & He, 2025) of the same scale using the same network achieves 76.7%. Random guessing yields 33.3%.

Table 3: Classification accuracy (%) when including original images as a fourth class.

| | Total | Claude-3.5-Sonnet | Gemini-1.5-Pro | GPT-4o | Original |
|---|---|---|---|---|---|
| Accuracy | 51.84 | 50.83 | 56.31 | 38.30 | 82.11 |

Table 4: Classification accuracy (%) with fixed SGD classifier.

| | Flux-schnell | SDXL | SD 2.1 | SD 1.5 |
|---|---|---|---|---|
| Accuracy | 46.05 | 45.69 | 44.76 | 41.67 |

## 4.2 CAPTION FINGERPRINTS FAIL TO TRANSFER INTO IMAGES

Despite the near-perfect attribution observed for captions, classification on generated images is far less successful. As shown in Figure 3, the best-performing model (Flux-schnell) reaches only 49.9% accuracy, barely above random guessing (33.3%) and well below the 76.7% accuracy achieved on natural images of similar scale (Liu & He, 2025). This indicates that the distinctive linguistic fingerprints of captions largely vanish once translated into the visual domain.

## 4.3 ABLATION STUDY

To better understand this discrepancy and further validate this finding, we performed ablation studies on top of the initial experiments.

**Add original images into classification.** To test whether the observed gap also holds against natural data, we introduced the 10k original images used in generating captions into the classification as the fourth class for direct comparison. The accuracies by class are listed in Table 3. Among the generated images, Gemini-1.5-Pro images are slightly more distinguishable (56.3%), while GPT-4o images are harder to classify (38.3%). The original images achieve 82.1%, highlighting a clear and consistent gap in idiosyncrasy between natural and generated data.

**Classification on extracted feature.** To examine whether our findings are sensitive to the classifier design, we trained a simple SGD classifier on CLIP features with fixed hyperparameters across different generation models and repeated runs. The test accuracies were consistently low (41.7–46.0%), as summarized in Table 4. This confirms that the difficulty in distinguishing generated images is not an artifact of classifier choice, feature representation, or training instability, but rather reflects the intrinsic similarity among generated samples.

## 5 THE IDIOSYNCRATIC GAP BETWEEN IMAGE CAPTIONING AND GENERATION MODELS

The classification results raise a key question: *Why are captions from different MLLMs easily distinguishable, while the corresponding generated images are not?*

Intuitively, if the distinctive tokens in captions were faithfully mapped into the visual modality, their signatures should also appear in generated images. Moreover, given the vast pixel space and color range available, images should in principle be capable of encoding more information than a short caption. The fact that this transfer fails suggests that some informative features are lost during generation—whether they are genuinely valuable or merely stylistic preferences.

### 5.1 LINGUISTIC ANALYSIS ON THE CAPTIONS

To test whether attribution relies on superficial cues, we modified and paraphrased captions before re-running classification. Simple edits included removing formatting, deleting special characters, or shuffling words and letters. In addition, we generated paraphrases using Qwen2.5 (Yang et al., 2024) with multiple prompts.

As shown in Table 5, idiosyncrasies lie primarily at the word level rather than individual characters, consistent with prior findings on LLMs (Sun et al., 2025). Even after paraphrasing, classification accuracy remains above 90%, confirming that model-specific signals are rooted not in surface form but in deeper factors such as descriptive perspective and content selection. Further qualitative analyses (See Appendix B) indicate that Claude-3.5-Sonnet adopts a narrative, context-oriented framing, Gemini-1.5-Pro emphasizes camera perspectives and exhaustive detail, and GPT-4o provides concise summaries focusing on salient objects and layout. These differences motivate a closer examination of how models represent visual content.

Table 5: Total and per-class accuracy under different text modifications and paraphrases. Paraphrasing was performed with three distinct prompts on `Qwen-2.5-1.5B-Instruct` and `Qwen-2.5-7B-Instruct` to ensure robustness across rewording styles and model scales. Detailed information about prompts is provided in Appendix C.

| Text Transformation | Total | Claude-3.5-Sonnet | Gemini-1.5-Pro | GPT-4o |
|---|---|---|---|---|
| Removing Markdown Format | 99.71 | 99.73 | 99.62 | 99.77 |
| Removing Special Characters | 99.78 | 99.78 | 99.78 | 99.77 |
| Shuffling Words | 99.42 | 99.43 | 99.60 | 99.23 |
| Shuffling Letters | 34.49 | 0.00 | 100.00 | 3.48 |
| Paraphrase 1 (Qwen-2.5-1.5B-Instruct) | 95.59 | 94.35 | 95.45 | 97.95 |
| Paraphrase 1 (Qwen-2.5-7B-Instruct) | 95.90 | 92.68 | 97.73 | 97.30 |
| Paraphrase 2 (Qwen-2.5-1.5B-Instruct) | 97.28 | 95.90 | 97.78 | 98.17 |
| Paraphrase 2 (Qwen-2.5-7B-Instruct) | 97.90 | 96.43 | 99.10 | 98.17 |
| Paraphrase 3 (Qwen-2.5-1.5B-Instruct) | 96.31 | 94.87 | 96.50 | 97.57 |
| Paraphrase 3 (Qwen-2.5-7B-Instruct) | 95.81 | 90.97 | 98.47 | 98.02 |

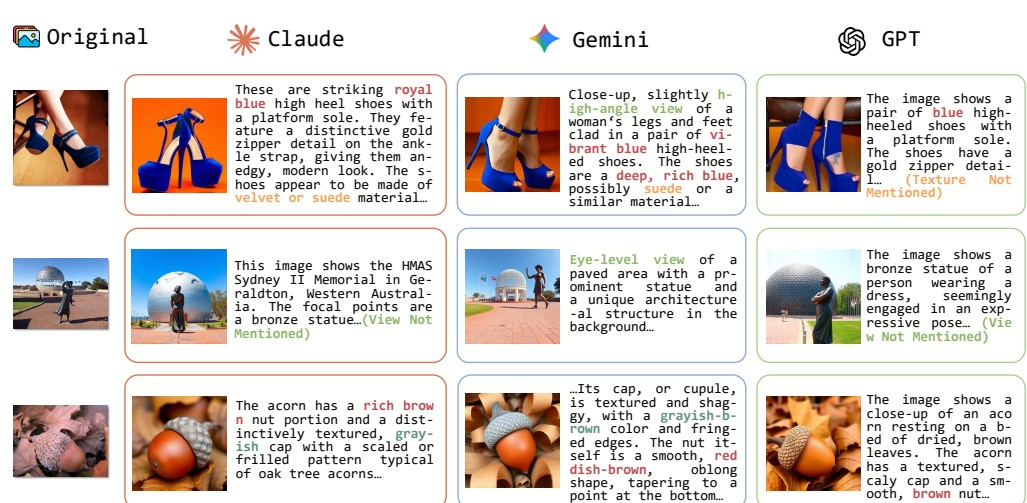

Figure 4: Comparisons of captions generated by `Claude-3.5-Sonnet`, `Gemini-1.5-Pro`, and `GPT-4o` on the same images. Each row shows one original image (left) and the corresponding model outputs.

## 5.2 VISUAL CONTENT ANALYSIS

By synthesizing these analyses and inspecting the generated data, we identify idiosyncrasies in four key dimensions: the level of descriptive detail, the use of color vocabulary, the use of texture vocabulary, and the treatment of compositional structure. We analyze each aspect in depth below.

**Level of descriptive detail.** We use `Qwen2.5-7B-Instruct` as a ranking model to order captions from most to least detailed, where "detail" is defined as the amount of specific, factual, and descriptive information provided. For each prompt, the three captions (from `Claude-3.5-Sonnet`, `Gemini-1.5-Pro`, and `GPT-4o`) are randomly shuffled and anonymized before evaluation to avoid bias. As shown in Fig. 5a, `Gemini-1.5-Pro` stands out in this category, with 84.27% of its captions judged most detailed. `GPT-4o` is ranked last in 71.96% of cases, while `Claude-3.5-Sonnet` most often occupies the middle rank (59.16%). These results reveal a clear hierarchy of descriptive richness across models: `Gemini` > `Claude` > `GPT`.

Similarly, on the generated images, we apply `GPT-4o` as the judge to rank the order on the level of details for the generated images. We randomly evaluate 1000 groups of images, which are also shuffled and anonymized before evaluation. As shown in Table 6, however, the rank is completely reversed. The richness of the details for the generated images is almost on the same level while the images from `GPT-4o` generated captions slightly outperforms the other two models. That concludes, the richness of the captions is not faithfully retained in the synthesized images.

Table 6: Detail ranking distribution of generated images. Rank 1, 2, and 3 denote the frequency with which each model was judged most, middle, or least detailed, respectively. Out of 1000 groups, 983 were successfully ranked, while 17 were ambiguous and could not be clearly assigned.

| Model | Rank 1 | Rank 2 | Rank 3 |
|---|---|---|---|
| Claude-3.5-Sonnet | 317 | 296 | 370 |
| Gemini-1.5-Pro | 282 | 346 | 355 |
| GPT-4o | 384 | 341 | 258 |

Table 7: Lexical statistics of color and texture vocabulary in model-generated captions (30k captions per model). Metrics include frequency of basic and nuanced vocabulary, proportion of captions containing at least one such term, and average frequency per caption. Percentages indicate the share of captions containing at least one occurrence, while averages are computed as total frequency divided by the number of captions.

| Model | Basic (Total) | Nuanced (Total) | With Basic (%) | With Nuanced (%) | Basic (Average) | Nuanced (Average) |
|---|---|---|---|---|---|---|
| *Color Vocabulary* | | | | | | |
| Claude-3.5-Sonnet | 88,797 | 23,235 | 92.36 | 43.96 | 2.96 | 0.77 |
| Gemini-1.5-Pro | 155,363 | 38,495 | 97.45 | 55.47 | 5.18 | 1.28 |
| GPT-4o | 62,843 | 10,186 | 81.01 | 22.95 | 2.09 | 0.34 |
| *Texture Vocabulary* | | | | | | |
| Claude-3.5-Sonnet | 27,009 | 32,034 | 67.24 | 52.91 | 0.90 | 1.07 |
| Gemini-1.5-Pro | 35,269 | 49,296 | 73.00 | 64.10 | 1.18 | 1.64 |
| GPT-4o | 27,859 | 26,862 | 67.67 | 50.83 | 0.93 | 0.90 |

**Color vocabulary.** We quantify the use of color terms with a deterministic dictionary-based matcher applied to normalized captions, counting both basic colors (e.g., red, green, blue) and nuanced variants (e.g., CSS/X11 shades, multi-word forms, and shade modifiers). As shown in Table 7, `Gemini-1.5-Pro` shows the highest frequency and coverage of color terms, `Claude-3.5-Sonnet` is similar but employs a slightly broader nuanced vocabulary, and `GPT-4o` uses color terms least often with the narrowest nuanced set. Yet these pronounced textual gaps do not yield proportionate separability in the image domain (Fig. 5b), suggesting that nuanced color instructions are often normalized away by T2I models.

**Texture vocabulary.** We assess captions for the use of texture descriptors, distinguishing between basic tactile terms (e.g., rough, smooth) and more nuanced expressions for materials, finishes, or fine-grained surface qualities. Judgments are made using `Qwen2.5-7B-Instruct`. As shown in Table 7, `Gemini-1.5-Pro` employs texture vocabulary most extensively, particularly nuanced terms, while `Claude-3.5-Sonnet` and `GPT-4o` use fewer such descriptors. Overall, Gemini demonstrates the richest texture lexicon, Claude is moderate, and GPT-4o the most limited. Again, we do not see a matching ordering in image attribution (Fig. 5b), consistent with the hypothesis that fine-grained material cues in captions are weakly realized by current T2I systems.

**Visual composition.** We perform a semantic analysis of captions to assess whether they encode key principles of photographic composition. Using `Qwen2.5-1.5B-Instruct`, we evaluate each caption against four criteria: (1) explicit description of spatial layers (foreground, middle ground, background), (2) identification of a main subject and its focus state, (3) mention of guiding elements such as leading lines or framing, and (4) reference to balance, symmetry, or subject placement.

Across 90,000 captions from 3 models, we observe clear differences in compositional awareness (Table 8). `Claude-3.5-Sonnet` consistently attains the highest coverage across all four criteria. In contrast, `Gemini-1.5-Pro` and `GPT-4o` score slightly lower on spatial layering and subject

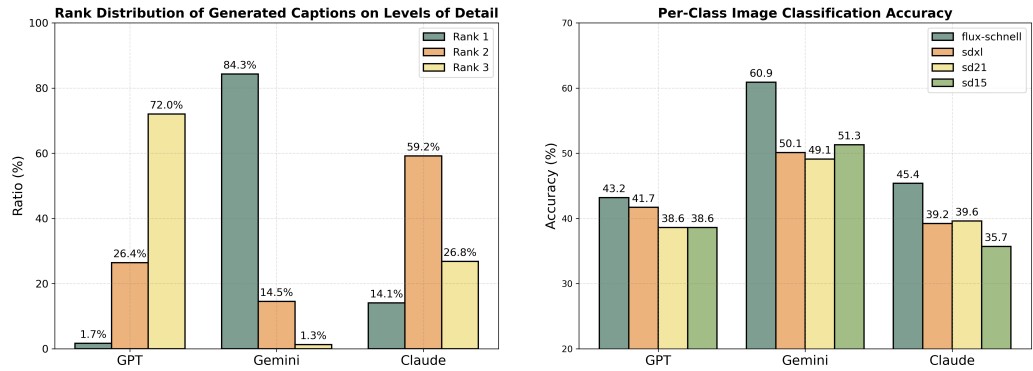

Figure 5: Comparison of captioning models: **(a)** detail ranking distribution of generated captions, and **(b)** per-class classification accuracy of generated images.

focus, and substantially lower on guiding elements and symmetry. Crucially, heightened compositional explicitness in text does not manifest as stronger per-class separability in images (Fig. 5b), implying that composition-related instructions are partially lost or regularized by the generator.

Table 8: Semantic composition analysis of captions. Values indicate the percentage of captions meeting each criterion.

| Model | Spatial Layers | Subject Focus | Guiding Elements | Balance / Symmetry |
|---|---|---|---|---|
| Claude-3.5-Sonnet | 93.38 | 96.55 | 86.67 | 3.52 |
| Gemini-1.5-Pro | 90.70 | 90.09 | 66.99 | 0.54 |
| GPT-4o | 86.42 | 88.61 | 70.48 | 1.07 |

## 6 DISCUSSION

In this work, we present systematic evidence of a clear gap between image captioning and generation models. We show through extensive experiments that the source model of a caption can be identified with near-perfect accuracy from text alone. However, these model-specific fingerprints disappear once captions are translated into images by current generators. Analyses of lexical, structural, and content-selection features suggest that the gap stems not from surface phrasing, but from deeper descriptive choices that are inconsistently reflected in images.

From the previous analysis, we find that image generation models often fail to preserve fine-grained caption details, a key factor in the idiosyncratic gap between captioning and generation. As shown in 5.2, the level of detail in captions is not fully reflected in the generated images. Nuanced color descriptions, for instance, rarely affect how colors are rendered. For composition, generation models may default to common scene structures based on context rather than strictly following specific words. These results highlight areas where current generation models can improve in the future.

ETHICS STATEMENT

This work investigates model-specific idiosyncrasies in image captioning and generation systems. Our study uses only publicly available datasets (CC3M, COCO, ImageNet, MNIST) and API-based model outputs, without involving human subjects or sensitive data.

REPRODUCIBILITY STATEMENT

We ensure reproducibility by providing detailed descriptions of datasets, models, prompts, and training configurations. All datasets used are standard and publicly accessible. Caption generation follows fixed prompts across three proprietary MLLMs (`Claude 3.5 Sonnet`, `Gemini 1.5 Pro`, `GPT-4o`), accessed via official APIs. Classifiers are trained with clearly specified hyperparameters, architectures, and evaluation splits. We will release code to replicate all experiments upon publication.

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

## A    LLM USABLE STATEMENT

Large language models were used in this work to assist with writing and editing tasks, such as polishing grammar, improving clarity, and suggesting alternative phrasings for sections of the manuscript. No LLM outputs were used as scientific claims, experimental findings, or conclusions. The authors take full responsibility for the accuracy and integrity of all content presented in this paper.

## B    LLM ANALYSIS ON THE CHARACTERISTICS OF THE GENERATED CAPTIONS

This part contains the complete text from three different large language models tasked with analyzing the distinctive features of captions generated by Claude-3.5-Sonnet, Gemini-1.5-Pro, and GPT-4o. The text is presented as originally generated, with only minor typographical edits to remove non-standard characters (e.g., emojis) for display compatibility.

### B.1    ANALYSIS FROM "CLAUDE-SONNET 4"

Looking at the captioning outputs from these three vision language models across 10,000 images, I can identify several distinctive language patterns for each:

#### B.1.1    CLAUDE (CLAUDE-3-5-SONNET)

**Distinctive Language Features:**

- Conversational and natural tone: Uses phrases like "This image shows..." or "In this image, we can see..."
- Concise yet descriptive: Provides good detail without being overly verbose.
- Contextual inference: Often adds interpretive context (e.g., "appears to be part of an organized event").
- Practical descriptions: Focuses on what's practically happening in scenes.
- Moderate technical vocabulary: Uses specific terms when appropriate but remains accessible.
- Smooth narrative flow: Descriptions read naturally without feeling mechanical.

**Identifying Markers:**

- Opening phrases: "This image shows...", "This is a..."
- Contextual additions: "suggesting...", "appears to be...", "likely..."
- Balanced detail level - not too sparse, not exhaustively detailed.
- Natural sentence structure and flow.

#### B.1.2    GEMINI (GEMINI-1.5-PRO)

**Distinctive Language Features:**

- Highly structured format: Consistently uses viewpoint descriptions ("Close-up, slightly low-angle view...").
- Extremely detailed and systematic: Provides exhaustive catalogs of visible elements.
- Technical precision: Uses specific photography/cinematography terms.
- Organized presentation: Often uses bullet points or structured breakdowns.
- Spatial awareness: Detailed positioning and relationship descriptions.
- Comprehensive coverage: Attempts to describe virtually every visible element.

**Identifying Markers:**

- Signature opening: Almost always starts with viewing angle/perspective description.
- Systematic enumeration: Lists elements methodically.
- Technical terminology: "Eye-level view", "low-angle", "close-up", etc.
- Spatial precision: Detailed positional relationships.
- Exhaustive detail: Much longer descriptions than other models.
- Structured formatting: Uses line breaks and organized sections.

### B.1.3 GPT-4O (GPT-4O)

**Distinctive Language Features:**

- Balanced and measured: Moderate detail level between Claude's brevity and Gemini's exhaustiveness.
- Slightly more formal tone: Professional but not overly technical.
- Thematic focus: Often emphasizes the overall scene or atmosphere.
- Descriptive but efficient: Good coverage without excessive detail.
- Clear and straightforward: Direct descriptions without much interpretive context.
- Consistent structure: Reliable format across different image types.

**Identifying Markers:**

- Standard opening: "The image shows..." or "The image depicts..."
- Moderate detail level: More detailed than Claude, less exhaustive than Gemini.
- Atmospheric descriptions: Often mentions mood, lighting, or overall impression.
- Balanced coverage: Describes main elements without getting lost in minutiae.
- Professional tone: Formal but accessible language.
- Consistent length: Descriptions tend to be similar in scope across images.

**Key Distinguishing Patterns:**

- **Length:** Gemini > GPT-4o > Claude
- **Technical detail:** Gemini > GPT-4o > Claude
- **Conversational tone:** Claude > GPT-4o > Gemini
- **Structural consistency:** Gemini > GPT-4o > Claude
- **Interpretive context:** Claude > GPT-4o > Gemini

**Most Reliable Identifiers:**

- **Gemini:** Look for viewing angle descriptions at the start and systematic, exhaustive detail.
- **Claude:** Look for conversational tone, contextual inference, and natural flow.
- **GPT-4o:** Look for "The image shows..." openings and balanced, professional descriptions.

Based on my analysis of the captioning samples, each model has very distinct linguistic fingerprints that make them quite identifiable: Gemini is the easiest to spot - it almost invariably opens with precise viewpoint descriptions like "Close-up, slightly low-angle view of..." and provides extremely systematic, exhaustive detail. It reads like a professional photography or cinematography analysis. Claude has a conversational, natural tone with contextual interpretation. It tends to be concise but adds meaningful context about what might be happening or the purpose of scenes. GPT-4o sits in the middle ground with professional but accessible language, moderate detail levels, and tends to emphasize overall atmosphere or themes. The most reliable identifying feature is probably Gemini's signature opening format describing camera angles and viewing perspectives, which appears consistently across all its samples. Claude's interpretive context-adding ("appears to be," "suggesting") and GPT-4o's balanced, thematic approach are also quite distinctive.

### B.2 ANALYSIS FROM "GPT-5"

Here's a synthesis of the distinctive language identities of the three captioning models (Claude-3.5-Sonnet, Gemini-1.5-Pro, and GPT-4o), based on the multi-example outputs you provided. I'll break this into model-by-model profiles, then give you practical "fingerprints" that can help you recognize each model in unseen captions.

#### B.2.1 CLAUDE-3.5-SONNET

**Stylistic Features:**

- **Concise but polished:** Usually 1–2 medium paragraphs, not overly long. Sentences are smooth and descriptive, but not overloaded with details.

- **Interpretive layer:** Often adds contextual or interpretive remarks (e.g., "suggests this is likely part of a European palace", "creating a humorous modern zombie moment", "perfect for a special occasion").

- **Elegant phrasing:** Uses words like "captured perfectly," "haunting," "tranquil," "reminiscent of". Feels almost like editorial writing or a photo-essay caption.

- **Specific but not hyper-granular:** Names obvious objects but rarely catalogs every tiny detail (compared to Gemini).

- **Balanced register:** Mixes factual description with mood/atmosphere, often noting lighting, ambience, or historical/cultural context.

**Identities:**

- Recognizable by interpretive commentary + smooth, compact prose.

- Frequent patterns: "This image shows. . . ", "The scene captures. . . ", "The overall aesthetic suggests. . . ".

- Captions read like professional magazine blurbs, not technical listings.

#### B.2.2 GEMINI-1.5-PRO

**Stylistic Features:**

- **Extremely detailed & structured:** Often breaks descriptions into bullet-style lists or sections ("Here's a breakdown of the visible elements").

- **Low-angle / high-angle qualifiers:** Very consistent in noting camera perspective ("eye-level view," "slightly low-angle," "close-up, over-the-shoulder").

- **Meticulous object listing:** Names minutiae others ignore (e.g., "a red tie holds scallions," "a skull and crossbones forward of the name," "a diamond-shaped pesticide sign").

- **Color & texture obsession:** Adds precise qualifiers like "muted grey-blue," "light brown potatoes," "bubble canopy," "murky brown water".

- **Long, multi-paragraph outputs:** Typically 3–5 dense blocks, making it the most verbose of the three.

**Identities:**

- Look for explicit scene geometry, camera angles, exhaustive object breakdowns, fine-grained texture/color.

- Common intro: "Eye-level view of. . . ", "Close-up, slightly low-angle view. . . ".

- Tone feels technical + photographic, almost like a field notebook or art catalog.

### B.2.3 GPT-4O

**Stylistic Features:**

- **Compact and plain:** Shortest answers overall, usually one paragraph (sometimes two).
- **Straightforward object naming:** Lists major entities but skips minor details.
- **Mild contextualization:** Notes obvious atmospheres ("festive," "tranquil," "theatrical"), but less interpretive than Claude.
- **Generalized adjectives:** Uses words like "lively," "picturesque," "ornate," "beautiful" without the subtle nuance Claude uses.
- **Dataset-savvy phrasing:** Occasionally references blurriness, motion, or resemblance to ML datasets ("reminiscent of MNIST digits," "slightly blurry, indicating movement").

**Identities:**

- Recognizable by brevity + general language.
- Captions feel functional, like "answering the prompt efficiently" rather than telling a story.
- Common phrasing: "The image shows. . .", "The scene features. . .", "In the background. . .".

**Quick Fingerprints (How to Spot Them Fast):**

- **Claude-3.5-Sonnet** → Interpretive + elegant, mid-length, adds atmosphere/context (sounds like a curated photo caption).
- **Gemini-1.5-Pro** → Verbose + hyper-detailed, mentions camera angle/lighting/textures, sometimes bullet-point structure.
- **GPT-4o** → Short + simple, names essentials, minimal elaboration, sometimes dataset-aware phrasing.

Would you like me to build a reference table with "tell-tale phrases/structures" (almost like linguistic fingerprints) so you can quickly identify which model likely produced a caption just by scanning it? That could help if you're planning classification or human evaluation.

## B.3 ANALYSIS FROM "GEMINI-2.5 PRO"

Based on the provided captioning outputs, each vision-language model demonstrates a distinct and recognizable linguistic style.

### B.3.1 CLAUDE 3.5 SONNET: THE NARRATIVE STORYTELLER

Claude's captions are characterized by their interpretive and narrative quality. It often goes beyond a simple description to infer the context, mood, or story behind the image. Its language is frequently evocative and focused on the overall atmosphere.

**Distinctive Features & Identities:**

- **Atmospheric Language:** Uses descriptive adjectives that convey a feeling or mood, such as "dynamic quality," "intimate perspective," "magical atmosphere," or "mysterious and haunting appearance."
- **Contextual Inference:** Frequently makes educated guesses about the event or location, such as "a zombie walk or Halloween event," "a dog show," or "a classic Seattle holiday scene."
- **Narrative Flow:** Presents descriptions in a single, flowing prose paragraph that reads like a short story or a scene description.
- **Potential Weakness:** Can sometimes misinterpret abstract or non-standard images by trying to force a real-world context. For example, it incorrectly identified a handwritten digit from the MNIST dataset as a "basketball jersey."

### B.3.2 GEMINI 1.5 PRO: THE METHODICAL ANALYST

Gemini's style is highly structured, detailed, and analytical. Its primary goal appears to be an exhaustive and objective cataloging of the visual information, often organized in a systematic way that is easy to parse.

**Distinctive Features & Identities:**

- **Camera Perspective Opening:** A very strong identifier is that its captions almost always begin by stating the camera's angle and proximity, such as "Close-up, slightly low-angle view," "Eye-level view," or "Over-the-shoulder view."

- **Structured Formatting:** Frequently uses double newlines to separate paragraphs, creating distinct logical blocks. For detailed prompts, it may even use bulleted lists with bolded headings to break down every component in the image.

- **Exhaustive Detail:** Tends to produce the longest and most granular descriptions, mentioning specific details like skin tone, indistinct graphics on a shirt, or the material of an object (e.g., "matte finish" on scissor handles).

- **Objective Tone:** The language is typically encyclopedic and neutral, focusing on what is visually present rather than the emotional tone of the scene.

### B.3.3 GPT-4O: THE EFFICIENT SUMMARIZER

GPT-4o is defined by its conciseness and directness. It excels at providing a quick, accurate, and highly efficient summary of the image's content without unnecessary elaboration or stylistic flair.

**Distinctive Features & Identities:**

- **Brevity:** Its captions are consistently the shortest of the three models. It distills the scene to its most essential elements.

- **Direct-to-the-Point:** It avoids introductory framing (like Gemini's camera angles) and immediately states the main subject of the image.

- **Balanced Tone:** The tone is generally neutral and factual but can incorporate a touch of descriptive language (e.g., "tranquil and picturesque atmosphere") when it adds significant value.

- **High-Level Accuracy:** It demonstrates strong and accurate recognition, even with abstract subjects. For instance, it not only identified the handwritten digit correctly but also astutely noted its resemblance to the "MNIST dataset," showing deep contextual knowledge.

## C PARAPHRASE PROMPTS

These are the three distinct prompts to generate paraphrased versions of captions, applied with both Qwen-2.5-1.5B and Qwen-2.5-7B to ensure that the robustness analysis is not tied to a single paraphrase style.

**Prompt 1.**

```
{src text}
Paraphrase:
```

**Prompt 2.**

```
Paraphrase the following text while maintaining the semantic
meaning of the original text.
{src text}
Paraphrase:
```

**Prompt 3.**

```
Paraphrase the following text while maintaining the semantic
meaning of the original text.  Do not add explanations,
suggestions, or follow-up questions.  Only output the
paraphrased text.
{src text}
Paraphrase:
```

