# OpenReview forum: "Lost in Translation: On the Idiosyncratic Gap between Image Captioning and Generation Models"
_ICLR.cc/2026/Conference — ICLR 2026 Conference Withdrawn Submission_

### Official Review · Reviewer_5C2U · 2025-10-28

**Soundness:** 1
**Presentation:** 2
**Contribution:** 1
**Rating:** 2
**Confidence:** 3

**Summary:**

The paper finds that different captioning models have different idiosyncrasies in language, which can be identified by a classifier with near-perfect accuracy. However, when the captions generated by different captioning models are used to generate images through popular image generators (Flux, SD), the classifier has a much harder time differentiating through the generated image. The author believes there is information lost in the generator model.

**Strengths:**

The paper provided a detailed analysis of the idiosyncrasies of different captioning models.

**Weaknesses:**

**The hypothesis is not sound.**

I disagree with the hypothesis that the generated images should also be easily identifiable; otherwise, the prompt information is lost. Specifically, recent image generative models are trained on re-captioned image data. Many employ multiple captioning models, so the image generator has seen different captioning styles describing the same image during training. Therefore, when conditioned on prompts with different captioning styles, the model should actually be expected to generate the same distribution of training images.

This does not indicate information lost. Ultimately, the captions are generated by different captioning models describing the same source images, and the generator recreates similar images invariant to the different captioning styles. This is actually a desirable thing, as the model is invariant to the different ways of captioning.

**Lack of practical contributions**

The paper only analyzes the different styles of the captioning model but does not propose anything useful that improves the generation quality. This contributes very little to practical use.

**Questions:**

I would like to hear the author's response to the weaknesses regarding the whole hypothesis of the paper.

---

### Official Review · Reviewer_oRnf · 2025-10-30

**Soundness:** 3
**Presentation:** 3
**Contribution:** 2
**Rating:** 6
**Confidence:** 4

**Summary:**

This paper presents an empirical analysis of the gap between the captions that describe an image, and the image that can be generated by those captions. The first step of the analysis shows that the captions generated by Sonnet 3.5, GPT4o, and Gemini 1.5 Pro can be easily distinguished from each other using a simple BERT classification model. However, using those very captions as the inputs to Stable Diffusion v1.5, v2.1, XL, and FLUX.1-schnell does not result in such a clear distinction in the generated data. The paper seeks to explain the reasons for why the captions are to readily distinguishable from each other, and why the images are not. This investigation includes an analysis of the types of adjectives used in the captions, and whether the captions use language that reflects visual composition. It was not clear about what the reader is supposed to take away from this paper, in terms of actionable tasks that could change this behaviour in image generation systems.

**Strengths:**

S1: The motivation for the work is clearly explained. This is an important and interesting problem in the Vision-Language space.

S2: The choice of models for caption generation and image generation is a good reflection of the state of the art. The experimental details are clearly explained, for the most part, e.g. how the models were prompted and how the classifiers were trained. There are some unresolved questions, see below.

**Weaknesses:**

W1: The caption generation models are restricted to API-based systems, whereas the image generation systems are open-weights models. This is a strange separation between It would have been interesting to include high-quality open-weights models, e.g. Gemma3 and Qwen3 in the analysis.

W2: L149. The decision to include MNIST images in the data that needs to be captioned does not make sense to me. What can we expected a captioning model to do when prompted to write increasingly detailed captions about a small image that depicts a single digit?

W3: The results throughout the paper do not show the difference in performance for the different types of captions generated for the images, or the opposite task. This makes it difficult to understand how different lengths of generated data might affect classifier performance.

W4: The caption analysis could be improved by adopting tooling from the literature, e.g. unique vocabulary coverage, syntactic complexity measures, TTR analysis [1, 2].

[1] Mitchell et al. A survey of current datasets for vision and language research. EMNLP 2015.
[2] van Miltenburg et al. Measuring the Diversity of Automatic Image Descriptions. COLING 2018.

**Questions:**

Q1: Which captions did you use as inputs for the image generation experiments? DiffusionDB [3] suggests that there is a clear length distribution preference from user-created examples for Stable Diffusion, and that the earlier versions of the models had a strict token length cut-off.

Q2: It was not clear from Section 4.1 if the ResNet-18 backbone was pretrained on ImageNet or if you directly trained it on the generated images. Can you clarify this?

Q3: Is the paper missing a Conclusion section? Section 6 Discussion does not really conclude the paper, in its current form.

[3] Wang et al. DIFFUSIONDB: A Large-scale Prompt Gallery Dataset for Text-to-Image Generative Models. ACL 2023.

---

### Official Review · Reviewer_mxjc · 2025-11-01

**Soundness:** 2
**Presentation:** 3
**Contribution:** 2
**Rating:** 2
**Confidence:** 4

**Summary:**

This paper asks whether model-specific “stylistic fingerprints” in MLLM-generated captions transfer to images produced by T2I systems. The authors train source-attribution classifiers on both text and image outputs and find near-perfect caption attribution but weak image attribution (≤50% with Flux). They further analyze lexical and structural differences (detail, color/texture vocabularies, composition cues) and argue that T2I models fail to realize these caption idiosyncrasies visually.

**Strengths:**

- Clear, well-scoped question and a simple, scalable experimental template (text-side vs image-side attribution).
- Image-side results are reported across multiple generators with reasonable training setups for the classifier, and the authors attempt several ablations.

**Weaknesses:**

- A central methodological issue is the role of the pretrained text encoders inside the T2I systems. SD1.5/2.1/SDXL all run captions through frozen CLIP/OpenCLIP encoders, and FLUX appears to use a CLIP-L/14 + T5 stack. This means much of the stylistic variation that distinguishes captioners could be normalized before it ever reaches the image generator (like UNet or DiT). The paper interprets the weak image-side attribution as a failure of “the generator to transfer idiosyncrasies,” but without decoupling encoder and generator, the evidence equally supports the simpler explanation that the encoder collapses style.

- Relatedly, sequence-length limits likely wash out differences for the SD family. CLIP-based pipelines typically cap inputs around 77 tokens, so the “detailed” and “very detailed” captions are routinely truncated, likely the parts where style, color, texture, and composition cues tend to accumulate. FLUX’s longer T5 context provides a partial counterexample and may explain its relatively higher image-side attribution in Fig. 3. Unless effective token lengths are controlled at the tokenizer level across models, the study risks conflating truncation effects with a putative cross-modal “translation gap.”

**Questions:**

- Could you add an encoder-side control? Specifically, instead of only training a BERT source classifier, take each caption, run it through a fixed CLIP ViT-L/14 text encoder (with its native tokenization and 77-token limit), extract the text embeddings, and do simple linear probing to predict the caption’s source. How does that accuracy (and calibration) compare to your BERT results under matched length controls? If CLIP embeddings alone recover most source information, that would suggest the encoder is where idiosyncrasies are preserved or collapsed; if CLIP largely erases them while BERT retains them, that would support your cross-modal gap claim. You might also include a parallel probe with T5 to assess the impact of sequence length.

- A more faithful design is to run the idiosyncrasy test on a unified multimodal models (UMM) that both captions and generates with the same tokenizer/encoder and shared internal representations. That removes cross-system confounds (tokenization quirks, sequence limits, encoder normalization) and asks the right question: do a model’s own caption-side idiosyncrasies survive its own image generation? If a gap still remains under this same-model setup, the conclusion of a genuine cross-modal translation failure would be far more compelling.

---

### Official Review · Reviewer_wud7 · 2025-11-03

**Soundness:** 1
**Presentation:** 3
**Contribution:** 1
**Rating:** 2
**Confidence:** 4

**Summary:**

This paper focuses on analyzing and understanding the idiosyncratic behaviours of the image captioning and generative models. Specifically, MLLM models can be identified using simple BERT-based caption classifiers. While, generated images from respective captions show minimal idiosyncratic behaviour that is we cannot identify the MLLM of the provided input prompt to the image generative model. This paper focuses on understanding the compositionality of the generated captions and attempts to propose this task as potential evaluation metric for T2I models.

**Strengths:**

* The paper is well written and easy to follow.
* The MLLM caption analysis is interesting and shows the types of biases each of these models observe.

**Weaknesses:**

* The entire hypothesis is flawed and no statistical guarantees are provided.
* It is not clear why we (as a community) should bridge this “translation” gap?
  - Ideally, it should be the opposite. MLLM captions should not have such signatures (which is mainly differences in understanding image as per the section 5 findings).
  - Additionally, there are well established benchmarks such as T2I-Compbench, DPG-Bench focusing on the T2I model evaluations and prompt following (including text to image to text as evaluation metric).
* Importantly, to understand the idiosyncratic behaviour, one needs to train the T2I models from scratch using only synthetic captions from different MLLMs independently. This will put important light on whether such signatures carry forwards to T2I models and does it affect their compositionality.

**Questions:**

* What is the caption classification performance with respect to different levels of details?
  - Do you see that short captions are easier to classify than long captions or opposite?

---

### Author Response · Authors · 2025-11-14

We thank the reviewers for their detailed and insightful feedback. We appreciate the reviewers’ recognition of our motivation, the clarity of our experimental setup, and the value of our text-side and image-side analyses. The raised points are highly constructive and align with our goal of more rigorously understanding caption-to-image causal effects. Below, we provide several important clarifications regarding the current scope of our work and address specific misunderstandings raised in the reviews.

Regarding the hypothesis, our quantitative results and qualitative analysis have illustrated semantic differences, not stylistic variations, in the captions generated by different models. Based on the observation that upstream captions differ in semantic content, our paper proposes to systematically investigate whether T2I models are able to generate images that reflect these differences.

We agree that training T2I models from scratch with isolated caption sources would provide the strongest causal evidence, and this is one of our main motivations for understanding the influence of using a single captioning model. However, this would require a tremendous computational cost. Our current work therefore focuses on a first-step diagnostic: whether semantic differences in captions leave detectable signals in the generated images, as an initial indication of this effect.

We also appreciate the suggestions on caption and length analysis. For the image-side evaluation, we collected full caption sets from GPT, Claude, and Gemini, and we now report classifier results using a ResNet-18 trained from scratch on the generated images. We will also add results from open-source captioners in the revised version.

---

### Note · Authors · 2025-11-14

I have read and agree with the venue's withdrawal policy on behalf of myself and my co-authors.